# Determinants of Quality of Life after Stroke in Southern Portugal: A Cross Sectional Community-Based Study

**DOI:** 10.3390/brainsci11111509

**Published:** 2021-11-14

**Authors:** Eva Lourenço, Mário Rui dos Mártires Sampaio, Hipólito Nzwalo, Emília Isabel Costa, José Luis Sánchez Ramos

**Affiliations:** 1Doctoral School, Huelva University, 21001 Huelva, Spain; 2Intensive Care Unit, Algarve University Hospital Center, 8000 Faro, Portugal; 3Nursing Department, Algarve University, 8000 Faro, Portugal; mariosampaio@hotmail.com (M.R.d.M.S.); eicosta@ualg.pt (E.I.C.); 4Tavira Health Center, Algarve Health Administration, 8800 Tavira, Portugal; 5Faculty of Medicine and Biomedical Sciences, Algarve University, 8000 Faro, Portugal; nzwalo@gmail.com; 6Stroke Unit, Algarve University Hospital Center, 8000 Faro, Portugal; 7Health Sciences Research Unit: Nursing, 3000 Coimbra, Portugal; 8Nursing Department, Huelva University, 21001 Huelva, Spain; jsanchez@uhu.es

**Keywords:** Quality of Life, health related quality of life, acute myocardial infarction, stroke

## Abstract

Introduction: the perception of Quality of Life (QoL) has been used to evaluate the treatment and evolution of several pathologies. QoL evaluation allows a better understanding of the patient and his adaptation to the disease. An observational, community-based and descriptive correlational study was carried out to analyze stroke survivors’ perception of QoL. Methods: consecutive case-series of stroke survivors (≥3 months) followed in a single public primary health center (“Tavira Primary Health Centre”) from Algarve, southern Portugal. The Portuguese version of the World Health Organization Quality of life instrument was administered in 102 stroke survivors. Results: Perception of QoL was associated (*p* < 0.05) with specific sociodemographic (age, sex, marital status, academic training), economic (monthly family income) and clinical factors (number of vascular risk factors, type of stroke, evolution, chronic mRankin score). On multivariate analysis, chronic mRankin score on physical (R^2^ = 0.406; F = 8.757; *p* < 0.001), psychological (R^2^ = 0.286; F = 5.536; *p* < 0.001) and general domain (R^2^ = 0.357; F = 7.287; *p* < 0.001); and family income (R^2^ = 0.160; F = 3.156; *p* < 0.005) on environmental domain, emerged as predictors of QoL. Conclusion: Different socio-demographic, economic and health factors are associated with post-stroke QoL. Description of QoL contributes to the overall evaluation of the impact of stroke on health and should be a priority for health professionals.

## 1. Introduction

Despite the decline of incidence and mortality rates [1], stroke remains a leading cause of morbidity worldwide [2]. Most studies addressing the health impact of stroke are based on short, long-term mortality and functional neurological outcome. However, even in patients without significant neurological disability, stroke can be associated with poor Quality of Life (QoL) [3,4,5,6]. QoL measurements are potentially more relevant and informative to patients than the evaluation of functional outcomes [7]. QoL evaluation may uncover insufficiently managed chronic health problems or facilitate a much broader description of stroke impact on survivors [8]. Worldwide, there are few studies addressing QoL among stroke survivors. These studies have identified factors such as functional status, depression, low socioeconomic status, or pain as the main determinants of QoL in stroke survivors [5,6,8,9,10,11]. Despite the extreme importance for patients and community, there are no community based published studies describing the QoL of stroke survivors in Portugal. Therefore, we sought to investigate which factors determine QoL of stroke survivors in a community representative population of Algarve, the southernmost part of Portugal.

## 2. Materials and Methods

### 2.1. Setting and Sample

A cross-sectional, descriptive study was carried out between February to September 2018 on consecutively presenting stroke survivors from a single public primary health care unit, the Tavira Health Centre in Tavira, Algarve. The centre serves a population of about 26,100 inhabitants [12]. Primary health care in Portugal is universally free and is the main place of regular patient follow-up for the majority of acute and chronic health conditions. The institutional electronic database was used to identify the patients meeting the following inclusion criteria: >18 years at the time of diagnosis; stroke diagnosed at least 3 months apart; followed at the Tavira Health Center. Patients were excluded in the presence of neurological deficits preventing effective communication and incomplete medical history. We used the World Health Organization (WHO) definition of stroke, “clinical syndrome of sudden onset of rapidly developing symptoms and signs of focal or global cerebral deficit with symptom lasting more than 24 h or leading to death with no apparent cause other than vascular origin” [13] confirmed by imaging.

The following variables were extracted from the electronic database: age, gender, marital status, formal education, employment status, income, existence of a caregiver, cardiovascular risk factors (hypertension, smoking, type 2 diabetes, sedentary lifestyle, obstructive apnea syndrome (OSAS), type 1 diabetes, alcoholism, atrial fibrillation, ischemic heart disease, dyslipidemia, obesity, hyperuricemia), functional neurological status (modified Rankin Scale or mRankin) [14]; stroke classification (“Oxfordshire Community Stroke Project” [15]-OCSP), previous medication and clinical evolution/sequelae and admission through “Vía Verde” or stroke code (a set of procedures aiming to rapidly provide acute reperfusion treatments in patients with stroke). 

### 2.2. Health-Related Quality of Life

Health related Quality of life (HRQoL) is a multidimensional medical study questionnaire developed by WHO. The WHOQoL-BREF (abbreviated) version 2.0 has 26 questions and was translated and validated to Portuguese language [16,17]. These 26 questions access the individual’s perceptions of their health and well-being over the previous two weeks. Responses to questions are on a 1-5 Likert scale where 1 represents “disagree” or “not at all” and 5 represents “completely agree” or “extremely”. Every six questions cover one of four domains: Physical, Psychological, Social, Environment. Two questions access a General domain. The WHOQoL-BREF questionnaire takes about 15 minutes to complete and was administered in a face-to-face interview.

### 2.3. Statistical Analysis

For descriptive and inferential statistics Statistical Package for the Social Sciences (SPSS), version 21.0, was used. Continuous variables were expressed as mean and standard deviation, and categorical variables were expressed as percentages. Linear association between quantitative variables was evaluated by Pearson correlation coefficient. Spearman correlation coefficient was used in case variables did not present normality and for ordinal variables. T test was applied to variables with normal distribution, in order to determine whether the results were statistically different. Non-parametric tests, such as the Mann–Whitney test (2 independent samples) and the Kruskal–Wallis test (>2 independent samples) were applied when one of the variables had less than 30 individuals or did not present normal distribution. For each domain of the WHOQoL-BREF logistic, regression analysis was used to identify predictors associated with the QoL of stroke survivors. The statistical significance of all tests was set as *p* < 0.05 by 2-tailed tests. The Algarve Regional Research Ethics Committee approved the study in accordance with the Helsinki Declaration of 1983 (CES 34/2018). All subjects provided informed consent.

## 3. Results

Of the 120 persons who met the inclusion criteria and could be located, 102 (85.8%) agreed to participate and completed the questionnaire. Table 1 and Table 2 resume, respectively, the sociodemographic and clinical characteristics of the patients included in the study. Table 1 shows that the majority (65/63.7%) were men and the mean age at the diagnosis of stroke was 67.7 years old (SD: 12.3). The mean time between the stroke event and the questionnaire was 29.52 months (SD: 32). Most of the patients were married (65/63.7%), had four years of schooling (55/53.9%), were retired (69/67.6%) and had an income lower than 580 euros (47/46.1%).

Ischemic stroke was the most common type (87/86.4%). The most prevalent risk factors at the time of the vascular event were hypertension (85/82.5%), dyslipidemia (61/59.2%) and sedentary lifestyle (51/49.5%). The proportion of stroke survivors with any functional neurological dependency was 10 (9.9%), 72 (69.5%) and 63 (61.6%), before stroke, immediately after stroke and at the time of the questionnaire (from December 2018 to June 2019), respectively (Table 2). The majority of patients had a single stroke (78/76.6%); 82 patients (80.4%) did not receive any acute reperfusion treatment; only 29 patients had a positive clinical evolution without neurological sequelae.

Table 3 and Table 4 resume the analysis of the association between specific domains of the HRQoL with different sociodemographic and clinical variables, respectively.

Higher age at the time of stroke onset was negatively correlated with the physical (*p* < 0.01) and social (*p* < 0.05) dimensions of the HRQoL. Divorced survivors had higher average scores on physical, psychological, social, and general dimensions than married (*p* < 0.05) or widowed survivors (*p* < 0.01). Survivors with more years of school (*p* < 0.01) and higher family income (*p* < 0.05) had higher perception in all dimensions of the HRQoL. On the contrary, retired survivors (*p* < 0.05) and those with current need of caregivers (*p* < 0.01) had worse perception in multidomain of the HRQoL (Table 3).

Survivors who were taking medication or with comorbidities had worse perception of HRQoL in the physical, psychological, social, and general domains (*p* < 0.01) (Table 4). In comparison to ischemic stroke survivors, patients who suffered a hemorrhagic stroke had worse perception in almost all dimensions of the HRQoL (*p* < 0.01).

On multivariate analysis (Table 5, Table 6, Table 7 and Table 8) chronic mRS was the only predictor found for the physical domain (R^2^ = 0.406; F = 8.757; *p* < 0.001), the psychological domain (R^2^ = 0.286; F = 5.536; *p* < 0.001) and the general domain (R^2^ = 0.357; F = 7.287; *p* < 0.001). Chronic mRS explained 40.6%, 28.6% and 35.7% of the variance, respectively. For the environmental domain, monthly family income emerged as a predictor, explaining 6% of the variance (R^2^ = 0.160; F = 3.156; *p* < 0.005).

## 4. Discussion

This is the first community-based study describing the QoL of stroke survivors in our country. The study demonstrated that in stroke survivors, several sociodemographic factors (being older, widow, less educated), economic factors (lower monthly family income) and clinical factors (hemorrhagic stroke type, comorbidities, high chronic mRankin score) are associated with reduced perception of QoL in different dimensions. These findings are similar to those reported in previous studies [5,6,8,9,10,11]. A large proportion of stroke survivors have poor perception of their QoL and probably face unmet long-term needs [6]. Sociodemographic, economic and health factors have been shown to influence the QoL of stroke survivors [18,19,20]. The relatively young age at the time of stroke, as well at the time of QoL evaluation, further emphasizes the potential for long-term impact and great repercussions to the survivors and consequently for the society [19,20]. QoL is very complex and dependent on several factors. For instance, in our study we have demonstrated that widowed survivors have poor QoL. However, widowed patients are generally older, at greater risk of functional decline and poor social support. In addition, the grieving process, per se, negatively impacts various aspects of physical and mental health that can lead to a decrease in the perception of QoL [19,20,21].

Indeed, among older patients and in patients with lower education, reduced HRQoL has been shown to be multifactorial [6,11]. Compliance with post-stroke management including rehabilitation, control of vascular risk factors; access to supporting networks within or outside the family helps improve the perception of QoL [8]. Low education and income influence the ability to control cardiovascular risk factors and the treatment of stroke, due to the difficulty in implementing preventive measures and problems in accessing information, treatment, and rehabilitation services [11]. Patients may encounter financial barriers to accessing high cost treatments, resulting in a cycle of health degradation and worse perception of QoL [6]. Additionally, the effort to put in place specific prevention strategies, including restrictive diets, adoption of healthy behaviors, and chronic use of medications—which are sometimes very expensive—can further cause low perception of QoL [6,8].

In our study, only chronic mRankin and income emerged as a predictor of poorer QoL. The mRankin scale is used to evaluate the functional neurological outcome after stroke and is one of the most widely used scales [11,22]. Therefore, it is expected that a higher mRankin score or higher dependency would be associated with a lower perception of QoL. Others have also demonstrated that patients with worse functional outcomes or more severe strokes have poor perception on some specific domains of HRQoL such as physical, psychological, and general health. These domains are highly associated with autonomy or dependence on their caregivers [8,22]. Improvement of post-stroke QoL demands an integrated view of each patient. Management of specific social or health problems, for instance, spasticity, can have a large impact in multiple domains of QoL [23].

The percentage of ischemic stroke versus haemorrhagic stroke is similar to that described in the medical literature, where it is reported that around 85–90% of all strokes are ischemic and the remaining are haemorrhagic strokes [24]. Despite that, haemorrhagic stroke is often more deadly than ischemic stroke. The justification may be related to the pathophysiology of the stroke itself, which may lead to a lower response of deficits to rehabilitation, more sequelae, greater dependence, and dysfunction, and consequently a worse perception of QoL [8,9,24].

There are several limitations to be considered. Depression, a major contributor for HQoL [3,16,18] was not specifically evaluated. The inclusion of mental health in the HRQoL partially attenuates this limitation. There is no consensus on when to evaluate the QoL of post-stroke patients. We evaluated patients with different post-stroke follow-up durations, which may have reduced the possibility of identifying meaningful differences. Nevertheless, in some studies, on the long-term follow-up no significant changes occurred in perception of QoL in stroke survivors [9,21]. Finally, the number of participants was small, preventing a more detailed and robust statistical analysis.

## 5. Conclusions

Stroke long term health impact constitutes an important problem for survivors, relatives, caregivers, and society. Measuring HRQoL in post-stroke patients represents an important contribution to the better understanding of the overall health status of stroke survivors and should be prioritized for health professionals [5].

Our study showed that the patient’s subjective wellbeing is influenced by social, economic and health factors. Further studies, with larger samples, are needed in order to analyse the relationship between QoL and different clinical and socio-economic factors.

## Figures and Tables

**Table 1 brainsci-11-01509-t001:** Sociodemographic characteristics of the stroke survivors (*n* = 102).

Sociodemographic Characteristics	Average	SD
Gender		
Age		
Current age	70.23	12.6
Age at stroke onset	67.75	12.3
Difference (months)	29.52	32
	N	%
Gender		
Male	65	63.7
Female	37	36.3
Marital status		
Single	6	5.9
Married/Unmarried couple	65	63.7
Widowed	21	20.6
Divorced/Separate	10	9.8
Formal education		
Can’t read or write	8	7.8
Never went to school, but knows how to read and write	12	11.8
4 years	55	53.9
6 years	1	1
9 years	12	11.8
12 years	8	7.8
Bachelor/Degree	6	5.9
Employment status		
Employed	18	17.6
Unemployed	3	2.9
Sick leave	12	11.8
Retired	69	67.6
Family income		
Less minimal wage	47	46.1
Between 1 and 2 minimal wages	42	41.2
3 or more minimal wages	13	12.7
Existence of caregiver?		
Yes	51	(50)
No	51	(50)

**Table 2 brainsci-11-01509-t002:** Clinical characteristics of the stroke survivors (*n* = 102).

Cardiovascular Risk Factor					N	%
Hypertension					85	82.5
Smoking					16	15.5
Sedentary lifestyle					51	49.5
Type 2 Diabetes Mellitus					32	31.1
Type 1 Diabetes Mellitus					5	4.9
Alcoholism					22	21.4
Atrial fibrillation					24	23.3
Ischemic heart disease					2	1.9
Dyslipidemia					61	59.2
Obesity					15	14.6
Stress					1	1
Hyperuricemia					18	17.5
Admission using “Vía Verde” or stroke code					15	14.7
Neurological Syndrome (OCSP)						
PACI					43	42.2
TACI					8	7.8
LACI					27	26.5
POCI					9	8.8
Not specified					1	1
Hemorrhagic stroke					14	13.7
**mRankin Scale**	**Before stroke**		**Hospital discharge**		**Chronic ***	
	**N**	**%**	**N**	**%**	**N**	**%**
No symptoms	92	90.2	30	29.4	39	38.2
No significant disability	7	6.9	12	11.8	16	15.7
Slight disability	1	1	17	15.7	19	18.6
Moderate disability	2	2	16	15.7	12	11.8
Moderate severe disability			23	22.5	13	12.7
Severe disability			4	3.9	3	2.9
Number of strokes			**N**		**%**	
1st episode			78		76.6	
2nd episode			17		16.7	
3rd or more episodes			7		6.9	
Treatment						
Fibrinolysis			11		10.8	
Conservative			82		80.4	
Surgery			2		2	
Stent angioplasty			4		3.9	
Thrombectomy			3		2.9	
Clinical evolution						
Positive without sequelae			29		28.4	
Negative with sequelae			73		71.6	

OCSP—Oxfordshire Community Stroke Project; TACI—total anterior circulation infarcts; PACI—partial anterior circulation infarcts; LACI—lacunar circulation infarcts; POCI—posterior circulation infarcts; ***** mRankin Scale at the time of the questionnaire application.

**Table 3 brainsci-11-01509-t003:** Association between HRQoL domains and sociodemographic variables (*n* = 102).

Variable	General	Physical	Psychological	Social	Environmental
Current age		R = −0.262		R = −0.247	
		*p* = 0.0081		*p* = 0.0121	
Civil Status	H = 12.891	H = 13.141	H = 17.099	H = 14.505	
	d.f. = 3	d.f. = 3	d.f. = 3	d.f. = 3	
	*p* = 0.0052	*p* = 0.0042	*p* = 0.0012	*p* = 0.0022	
Formal education	Rho = 0.261	Rho = 0.368	Rho = 0.337	Rho = 0.354	Rho = 0.272
	*p* = 0.0083	*p* = 0.0013	*p* = 0.0013	*p* = 0.0013	*p* = 0.0013
Family income	Rho = 0.272	Rho = 0.205	Rho = 0.261	Rho = 0.396	Rho = 0.272
	*p* = 0.0063	*p* = 0.0383	*p* = 0.0083	*p* = 0.0013	*p* = 0.0063
Retired	t = −2.147	t = −3.340	t = −2.652	t = −3.823	
	d.f. = 101	d.f. = 101	d.f. = 101	d.f. = 101	
	*p* = 0.0344	*p* = 0.0014	*p* = 0.0094	*p* = 0.0014	
Existence of a caregiver	t = −5.816	t = −6.890	t = −5.176	t = −3.192	
	d.f. = 101	d.f. = 101	d.f. = 101	d.f. = 101	
	*p* = 0.0014	*p* = 0.0014	*p* = 0.0014	*p* = 0.0024	

R: Pearson correlation; H: Kruskal–Wallis test; t-Student test; Rho: Spearmann correlation.

**Table 4 brainsci-11-01509-t004:** Association between HRQoL domains of the and clinical variables (*n* = 102).

Variable	General	Physical	Psychological	Social	Environmental
Nº vascular risk factors				R = −0.209	
				*p* = 0.0341	
No medication	U = 476.5	U = 426.5	U = 290.5	U = 439.5	
	*p* = 0.0062	*p* = 0.0022	*p* < 0.0012	*p* = 0.0022	
Without sequelae	U = 416.5			U = 504.0	
	*p* = 0.0052			*p* = 0.0402	
Discharge mRankin	Rho = −0.487	Rho = −0.458	Rho = −0.402	Rho = −0.285	
	*p* = 0.0013	*p* = 0.0013	*p* = 0.0013	*p* = 0.0033	
Chronic mRankin	Rho = −0.622	Rho = −0.629	Rho = −0.550	Rho = −0.314	
	*p* = 0.0013	*p* = 0.0013	*p* = 0.0013	*p* = 0.0013	
Haemorrhagic stroke	U = 259.5	U = 324.5	U = 303.5		
	*p* = 0.0012	*p* = 0.0042	*p* = 0.0022		
Clinical evolution	t = 6.133	t = 5.571	t = 5.472	t = 4.686	t = 2.679
	d.f. = 101	d.f. = 101	d.f. = 101	d.f. = 101	d.f. = 101
	*p* = 0.0014	*p* = 0.0014	*p* = 0.0014	*p* = 0.0014	*p* = 0.0094

R: Pearson correlation; U: Mann–Whitney test; t-Student test; Rho: Spearmann correlation.

**Table 5 brainsci-11-01509-t005:** Multivariate analysis for Physical domain.

Physical Domain	Unstandardized Coefficients		Standardized Coefficients	t	Sig.	95% Conf. Int.	
	B	Std. Error	Beta				
(Constant)	70.566	11.164		6.321	0.000	48.396	92.736
Current age	4.954	3.636	3.346	1.363	0.176	−2.266	12.174
Disease age	−5.054	3.640	−3.330	−1.388	0.168	−12.283	2.175
Difference (months)	−0.445	0.316	−0.761	−1.406	0.163	−1.073	0.184
Sex	−2.341	3.115	−0.061	−.751	0.454	−8.527	3.846
Family income	3.144	2.225	0.117	1.413	0.161	−1.274	7.562
Number of Cardiovascular Risk Factors	−1.175	1.084	−0.088	−1.085	0.281	−3.327	0.977
Previous mRankin	2.216	3.003	0.061	0.738	0.463	−3.748	8.180
Discharge mRankin	0.802	1.541	0.071	0.520	0.604	−2.259	3.863
Chronic mRankin	−7.761	1.674	−0.645	−4.637	0.000	−11.084	−4.438

**Table 6 brainsci-11-01509-t006:** Multivariate analysis for Psychological domain.

Psychological Domain	Unstandardized Coefficients		Standardized Coefficients	t	Sig.	95% Conf. Int.	
	B	Std. Error	Beta				
(Constant)	72.729	12.245		5.940	0.000	48.414	97.045
Current age	3.890	3.988	2-627	0.975	0.332	−4.029	11.809
Disease age	−3.836	3.993	−2.528	−0.961	0.339	−11.765	4.093
Difference (months)	−0.368	0.347	−0.630	−1.062	0.291	−1.057	0.320
Sex	−5.849	3.417	−0.152	−1.712	0.090	−12.634	0.936
Family income	0.601	2.440	0.022	0.246	0.806	−4.244	5.447
Number of Cardiovascular Risk Factors	−0.606	1.189	−0.045	−0.510	0.611	−2.966	1.754
Previous mRankin	−1.316	3.294	−0.036	−0.399	0.690	−7.857	5.225
Discharge mRankin	0.866	1.691	0.077	0.512	0.610	−2.491	4.223
Chronic mRankin	−7.175	1.836	−0.597	−3.909	0.000	−10.820	−3.530

**Table 7 brainsci-11-01509-t007:** Multivariate analysis for General domain.

General Domain	Unstandardized Coefficients		Standardized Coefficients	t	Sig.	95% Conf. Int.	
	B	Std. Error	Beta				
(Constant)	65.020	14.949		4-350	0.000	35.334	94.705
Current age	2.550	4.868	1.339	0.524	0.602	−7.118	12.217
Disease age	−2.445	4.874	−1.253	−0.502	0.617	−12.125	7.234
Difference (months)	−0.171	0.423	−0.227	−0.404	0.687	−1.012	0.670
Sex	−3.788	4.172	−0.076	−0.908	0.366	−12.072	4.496
Family income	−2.122	2.979	−0.061	−0.712	0.478	−8.038	3.794
Number of Cardiovascular Risk Factors	−0.741	1.451	−0.043	−0.511	0.611	−3.622	2.141
Previous mRankin	1.309	4.021	0.028	0.325	0.746	−6.677	9.294
Discharge mRankin	0.935	2.064	0.065	0.453	0.652	−3.163	5.034
Chronic mRankin	−11.108	2.241	−0.718	−4.957	0.000	−15.558	−6.658

**Table 8 brainsci-11-01509-t008:** Multivariate analysis for Environmental domain.

Environmental Domain	Unstandardized Coefficients		Standardized Coefficients	t	Sig.	95% Conf. Int.	
	B	Std. Error	Beta				
(Constant)	50.166	9.844		5.096	0.000	30.618	69.713
Current age	−0.305	3.206	−0.278	−0.095	0.924	−6.671	6.061
Disease age	0.246	3.210	0.219	0.077	0.939	−6.128	6.620
Difference (months)	0.128	0.279	0.296	0.459	0.647	−0.426	0.682
Sex	1.435	2.747	0.050	0.522	0.603	−4.020	6.890
Family income	7.393	1.962	0.371	3.769	0.000	3.497	11.288
Number of Cardiovascular Risk Factors	0.802	0.955	0.081	0.839	0.403	−1.096	2.699
Previous mRankin	−2.466	2.648	−0.092	−0.931	0.354	−7.725	2.792
Discharge mRankin	−1.002	1.359	−0.121	−0.737	0.463	−3.701	1.697
Chronic mRankin	−0.420	1.476	−0.047	−0.285	0.777	−3.350	2.510

## Data Availability

Authors agree to make data and materials supporting the results or analyses presented in their paper available upon reasonable request.

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
