# Peer review of "Determinants of Quality of Life after Stroke in Southern Portugal: A Cross Sectional Community-Based Study"

_brainsci, 2021, doi:10.3390/brainsci11111509_

Round 1

Reviewer 1 Report

I have no suggestions

Author Response

We thank you for having time to read our article.

Reviewer 2 Report

Quality of life is a significant measurement of the impact of stroke. Different QoL tools were used in the literature with different outcomes: quality of life by improving movement, daily activities, mental health, and muscle tone, for example. New outcomes also revealed this research:  https://www.ncbi.nlm.nih.gov/pmc/articles/PMC8303388/

Line 83: you can not apply t test if the variable is not normally distributed. You need to check first for the Gaussian distribution.

What did you mean by "current" (Table 2)? The time of the questionnaire (December 2018-June 2019)?

Table 3: why did you use Pearson correlation for age (why not for general, psychological and environmental) and Spearman correlation for formal education, family income? What is g.l.? You need to include these tests (also Kruskal-Wallis test) in the Methods section and describe when you used them better. The same with Table 4 - you need to redo the statistical analysis.

You said univariate analysis was done, but you presented only multivariate analysis in tables 4-8.

Line 165- delete "(Table 5)". You do not need to write it in Discussion section.

Also, you need to discuss more your results in the Discussion section. Start the paragraph with your results and compare them with the literature.

English needs extensive editing.

Author Response

We wish to thank you for your constructive comments. Your comments provided valuable insights to improve t manuscript.

Quality of life is a significant measurement of the impact of stroke. Different QoL tools were used in the literature with different outcomes: quality of life by improving movement, daily activities, mental health, and muscle tone, for example. New outcomes also revealed this research:  https://www.ncbi.nlm.nih.gov/pmc/articles/PMC8303388/

We thank the reviewer for pointing out this relevant manuscript. In fact, this manuscript contains relevant data to put in context some of our study findings.

“Improvement of post-stroke QoL demands demands an integrated view of each patient. Management  of specific social or health problems, for instance, spasticy, can have a large impact in multiple domains of QoL [23].”  Line 191-2

Line 83: you can not apply t test if the variable is not normally distributed. You need to check first for the Gaussian distribution.

We have, accordingly, modified the methods section to clarify this point:“Spearman correlation coefficient was used in case variables did not present normality and for ordinal variables. T test was applied to variables with normal distribution, in order to determine whether the results were statistically different. Non-parametric tests, as the Mann-Whitney test (2 independent samples) and the Kruskal-Wallis test (> 2 independent samples) were applied when one of the variables had less than 30 individuals or did not present normal distribution.” Line 87-93.

What did you mean by "current" (Table 2)? The time of the questionnaire (December 2018-June 2019)?

We thank the reviewer for catching this imprecision. We changed current by chronic with a note on the bottom of the table to clarify (mRankin Scale at the time of the questionnaire application).

Table 3: why did you use Pearson correlation for age (why not for general, psychological and environmental) and Spearman correlation for formal education, family income? What is g.l.? You need to include these tests (also Kruskal-Wallis test) in the Methods section and describe when you used them better. The same with Table 4 - you need to redo the statistical analysis.

We agree with the reviewer suggestion. We re-evaluate all the statistics to correct any inconsistency. The revised manuscript included a paragraph in the methods section to accommodate this point.

“Spearman correlation coefficient was used in case variables did not present normality and for ordinal variables. T test was applied to variables with normal distribution, in order to determine whether the results were statistically different. Non-parametric tests, as the Mann-Whitney test (2 independent samples) and the Kruskal-Wallis test (> 2 independent samples) were applied when one of the variables had less than 30 individuals or did not present normal distribution.” Line 87-95.

You said univariate analysis was done, but you presented only multivariate analysis in tables 4-8.

We understand the reviewer point. Table 3 and 4 are presenting the bivariate analysis.

Line 165- delete "(Table 5)". You do not need to write it in Discussion section.

Also, you need to discuss more your results in the Discussion section. Start the paragraph with your results and compare them with the literature.

We accept the reviewer suggestion. The following paragraph was added to accommodate this suggestion:

This is first community-based study describing the QoL of stroke survivors in our country.  The study demonstrated that in stroke survivors, several sociodemographic factors (being older, widow, less educated), economic factors (lower monthly family income) and clinical factors (hemorrhagic stroke type, comorbidities, high chronic mRankin score) are associated reduced perception of QoL in different dimensions. These findings are similar to those reported in previous studies [5,6,8-11]. Line 155-66.

English needs extensive editing.

I accept the reviewer suggestion.  A native English user revised the Manuscript. 

Reviewer 3 Report

Thank you for the opportunity to review this manuscript whose subject has been much studied, but it's still of interest.
The manuscript should be improved in the following sections:
Abstract: Provide a summary containing sections and especially the results. Remove the keyword: "Acute Myocardial Infarction" (there was only in two patients), and add "vascular risk factors" as a keyword.
Introduction: I recommend reviewing the topic and adding more recent evidence.
Results: In Tables 5,6,7,8 of the multivariate analyzes, add the 95% confidence interval.
In relation to the limitations, the participants had the stroke in a very different interval between them, it's a important methodological aspect that the authors comment on, and it makes difficult the comparison of their quality of life, the results should be interpreted with caution. In addition, the authors can explain factors associated with quality of life, not predictive factors.

Author Response

We wish to thank you for your constructive comments. Your comments provided valuable insights to improve t manuscript.

Thank you for the opportunity to review this manuscript whose subject has been much studied, but it's still of interest.
The manuscript should be improved in the following sections:
Abstract: Provide a summary containing sections and especially the results. Remove the keyword: "Acute Myocardial Infarction" (there was only in two patients), and add "vascular risk factors" as a keyword
.

We agree with the reviewer suggestion. The abstract now contains sections and the main results. “Vascular risk factors” was included as keyword.

Introduction: I recommend reviewing the topic and adding more recent evidence.
Results: In Tables 5,6,7,8 of the multivariate analyzes, add the 95% confidence interval.
In relation to the limitations, the participants had the stroke in a very different interval between them, it's a important methodological aspect that the authors comment on, and it makes difficult the comparison of their quality of life, the results should be interpreted with caution. In addition, the authors can explain factors associated with quality of life, not predictive factors.

We accept the reviewer suggestion. We included in the discussion section, some sentences to put our findings in perspective:

“The study demonstrated that in stroke survivors, several sociodemographic factors (being older, widow, less educated), economic factors (lower monthly family income) and clinical factors (hemorrhagic stroke type, comorbidities, high chronic mRankin score) are associated reduced perception of QoL in different dimensions. These findings are similar to those reported in previous studies [5,6,8-11]”.(line 157-61)

Improvement of post-stroke QoL demands demands an integrated view of each patient. Management  of specific social or health problems, for instance, spasticy, can have a large impact in multiple domains of QoL [23] (line 231-233)

All tables now contain the confidence interval.

We made clear in the text and in the abstract wich factors are just associated and which are predictive:

Perception of QoL was associated (p< 0.05) with specific sociodemographic (age, sex, marital status, academic training), economic (monthly family income) and clinical factors (number of vascular risk factors, type of stroke, evolution, chronic mRankin score). On multivariate analysis, chronic mRankin score on physical (R2= 0.406; F=8.757; p<0.001), psychological (R2= 0.286; F=5.536; p<0.001)  and general domain (R2= 0.357; F=7.287; p<0.001); and family income (R2= 0.160; F=3.156; p<0.005) on environmental domain, emerged as predictors of QoL”

We included the follow sentence to acknowledge  the limitation of different  follow-up duration:

“We evaluated patients with different post-stroke follow-up duration, which may have reduced the possibility of identification meaningful differences (line 244-45)”

Round 2

Reviewer 2 Report

The manuscript could be accepted in the present form after minor English editing.